# The Expxorcist: Nonparametric Graphical Models Via Conditional Exponential Densities

**Arun Sai Suggala** [*]
Carnegie Mellon University
Pittsburgh, PA 15213

**Mladen Kolar** [†]
University of Chicago
Chicago, IL 60637

**Pradeep Ravikumar** [‡]
Carnegie Mellon University
Pittsburgh, PA 15213

## Abstract

Non-parametric multivariate density estimation faces strong statistical and computational bottlenecks, and the more practical approaches impose near-parametric assumptions on the form of the density functions. In this paper, we leverage recent developments to propose a class of non-parametric models which have very attractive computational and statistical properties. Our approach relies on the simple function space assumption that the conditional distribution of each variable conditioned on the other variables has a non-parametric exponential family form.

## 1 Introduction

Let $X = (X_1, \ldots, X_p)$ be a $p$-dimensional random vector. Let $G = (V, E)$ be the graph that encodes conditional independence assumptions underlying the distribution of $X$, that is, each node of the graph corresponds to a component of vector $X$ and $(a, b) \in E$ if and only if $X_a \not\perp\!\!\!\perp X_b \mid X_{\neg ab}$ with $X_{\neg ab} := \{X_c \mid c \in V \setminus \{a, b\}\}$. The graphical model represented by $G$ is then the set of distributions over $X$ that satisfy the conditional independence assumptions specified by the graph $G$.

There has been a considerable line of work on learning *parametric* families of such graphical model distributions from data [22, 20, 13, 28], where the distribution is indexed by a finite-dimensional parameter vector. The goal of this paper, however, is on specifying and learning *nonparametric* families of graphical model distributions, indexed by infinite-dimensional parameters, and for which there has been comparatively limited work. *Non-parametric multivariate density estimation* broadly, even without the graphical model constraint, has not proved as popular in practical machine learning contexts, for both statistical and computational reasons. Loosely, estimating a non-parametric multivariate density, with mild assumptions, typically requires the number of samples to scale exponentially in the dimension $p$ of the data, which is infeasible even in the big-data era when $n$ is very large. And the resulting estimators are typically computationally expensive or intractable, for instance requiring repeated computations of multivariate integrals.

We present a review of multivariate density estimation, that is necessarily incomplete but sets up our proposed approach. A common approach dating back to [15] uses the logistic density transform to satisfy the unity and positivity constraints for densities, and considers densities of the form $f(X) = \frac{\exp(\eta(X))}{\int_{\mathcal{X}} \exp(\eta(x))dx}$, with some constraints on $\eta$ for identifiability such as $\eta(X_0) = 0$ for some $X_0 \in \mathcal{X}$ or $\int_{\mathcal{X}} \eta(x)dx = 0$.

With the logistic density transform, differing approaches for non-parametric density estimation can be contrasted in part by their assumptions on the infinite-dimensional function space domain of $\eta(\cdot)$. An early approach [8] considered function spaces of functions with bounded "roughness" functionals. The predominant line of work however has focused on the setting where $\eta(\cdot)$ lies in a Reproducing Kernel Hilbert Space (RKHS), dating back to [21]. Consider the estimation of these logistic density

---

[*]asuggala@cs.cmu.edu    [†]mkolar@chicagobooth.edu    [‡]pradeepr@cs.cmu.edu

transforms $\eta(X)$ given $n$ i.i.d. samples $\mathbb{X}_n = \{X^{(i)}\}_{i=1}^n$ drawn from $f_\eta(X)$. A natural loss functional is penalized log likelihood, with a penalty functional that ensures a smooth fit with respect to the function space domain: $\ell(\eta; \mathbb{X}_n) := -\frac{1}{n} \sum_{i \in [n]} \eta(X^{(i)}) + \log \int \exp(\eta(x)) dx + \lambda \operatorname{pen}(\eta)$, for functions $\eta(\cdot)$ that lie in an RKHS $\mathcal{H}$, and where $\operatorname{pen}(\eta) = \|\eta\|_\mathcal{H}^2$ is the squared RKHS norm. This was studied by many [21, 11, 6]. A crucial caveat is that the representer theorem for RKHSs does not hold. Nonetheless, one can consider finite-dimensional function space approximations consisting of the linear span of kernel functions evaluated at the sample points [12]. Computationally this still scales poorly with the dimension due to the need to compute multidimensional integrals of the form $\int \exp(\eta(x) dx$ which do not, in general, decompose. These approximations also do not come with strong statistical guarantees.

We briefly note that the function space assumption that $\eta(\cdot)$ lies in an RKHS could also be viewed from the lens of an infinite-dimensional exponential family [4]. Specifically, let $\mathcal{H}$ be a Reproducing Kernel Hilbert Space with reproducing kernel $k(\cdot, \cdot)$, and inner product $\langle \cdot, \cdot \rangle_\mathcal{H}$. Then $\eta(X) = \langle \theta(\cdot), k(X, \cdot) \rangle_\mathcal{H}$, so that the density $f(X)$ can in turn be viewed as a member of an infinite-dimensional exponential family with sufficient statistics $k(X, \cdot) : \mathcal{X} \mapsto \mathcal{H}$, and natural parameter $\theta(\cdot) \in \mathcal{H}$. Following this viewpoint, [4] propose estimators via linear span approximations similar to [11].

Due to the computational caveat with exact likelihood based functionals, a line of approaches have focused on penalized *surrogate* likelihoods instead. [14] study the following loss functional: $\ell(\eta; \mathbb{X}_n) := \frac{1}{n} \sum_{i \in [n]} \exp(-\eta(X^{(i)})) + \int \eta(x) \rho(x) dx + \lambda \operatorname{pen}(\eta)$, where $\rho(X)$ is some fixed known density with the same support as the unknown density $f(X)$. While this estimation procedure is much more computationally amenable than minimizing the exact penalized likelihood, the caveat, however, is that for a general RKHS this requires solving higher order integrals. The next level of simplification has thus focused on the form of the logistic transform function itself. There has been a line of work on an ANOVA type decomposition of the logistic density function into node-wise and pairwise terms: $\eta(X) = \sum_{s=1}^p \eta_s(X_s) + \sum_{s=1}^p \sum_{t=s+1}^p \eta_{st}(X_s, X_t)$. A line of work has coupled such a decomposition with the assumption that each of the terms lie in an RKHS. This does not immediately provide a computational benefit: with penalized likelihood based loss functionals, the loss functional does not necessarily decompose into such node and pairwise terms. [24] thus couple this ANOVA type pairwise decomposition with a score matching based objective. [10] use the above decomposition with the surrogate loss functional of [14] discussed above, but note that this still requires the aforementioned function space approximation as a linear span of kernel evaluations, as well as two-dimensional integrals.

A line of recent work has thus focused on further stringent assumptions on the density function space, by assuming some components of the logistic transform to be finite-dimensional. [30] use an ANOVA decomposition but assume the terms belong to finite-dimensional function spaces instead of RKHSs, specified by a pre-defined finite set of basis functions. [29] consider logistic transform functions $\eta(\cdot)$ that have the pairwise decomposition above, with a specific class of parametric pairwise functions $\beta_{st} X_s X_t$, and non-parametric node-wise functions. [17, 16] consider the problem of estimating monotonic node-wise functions such that the transformed random vector is multivariate Gaussian; which could also be viewed as estimating a Gaussian copula density.

To summarize the (necessarily incomplete) review above, non-parametric density estimation faces strong statistical and computational bottlenecks, and the more practical approaches impose stringent near-parametric assumptions on the form of the (logistic transform of the) density functions. In this paper, we leverage recent developments to propose a very computationally simple non-parametric density estimation algorithm, that still comes with strong statistical guarantees. Moreover, the density could be viewed as a graphical model distribution, with a corresponding sparse conditional independence graph.

Our approach relies on the following simple function space assumption: that the conditional distribution of each variable conditioned on the other variables has a non-parametric exponential family form. As we show, for there to exist a consistent joint density, the logistic density transform with respect to a particular base measure necessarily decomposes into the following semi-parametric form: $\eta(X) = \sum_{s=1}^p \theta_s B_s(X_s) + \sum_{s=1}^p \sum_{t=s+1}^p \theta_{st} B_s(X_s) B_t(X_t)$ in the pairwise case, with both a parametric component $\{\theta_s : s = 1, \ldots, p\}, \{\theta_{st} : s < t; s, t = 1, \ldots, p\}$, as well as non-parametric components $\{B_s : s = 1, \ldots, p\}$. We call this class of models the "expxorcist", fol-

lowing other "ghostbusting" semi-parametric models such as the nonparanormal and nonparanormal skeptic [17, 16].

Since the conditional distributions are exponential families, we show that there exist computationally amenable estimators, even in our more general non-parametric setting, where the sufficient statistics have to be estimated as well. The statistical analysis in our non-parametric setting however is more subtle, due in part to non-convexity and in part to the non-parametric setting. We also show how the Expxorcist class of densities is closely related to a semi-parametric exponential family copula density that generalizes the Gaussian copula density of [17, 16]. We corroborate the applicability of our class of models with experiments on synthetic and real data sets.

## 2  Multivariate Density Specification via Conditional Densities

We are interested in the approach of estimating a multivariate density by estimating node-conditional densities. Since node-conditional densities focus on the density of a single variable, though conditioned on the rest of the variables, estimating these is potentially a simpler problem, both statistically and computationally, than estimating the entire joint density itself. Let us consider the general non-parametric conditional density estimation problem. Given the general multivariate density $f(X) = \frac{\exp(\eta(X))}{\int_{\mathcal{X}} \exp(\eta(x))dx}$, the conditional density of a variable $X_s$ given the rest of the variables $X_{-s}$ is given by $f(X_s \mid X_{-s}) = \frac{\exp(\eta((X_s, X_{-s})))}{\int_{\mathcal{X}_s} \exp(\eta((x, X_{-s})))dx}$, which does not have a multi-dimensional integral, but otherwise does not have a computationally amenable form. There has been a line of work on such conditional density estimation, mirroring developments in multivariate density estimation [9, 18, 23], but unlike parametric settings, there are no large sample complexity gains with non-parametric conditional density estimation under general settings. There have also been efforts to use ANOVA decompositions in a conditional density context [31, 26].

In addition to computational and sample complexity caveats, recall that in our context, we would like to use conditional density estimates to infer a joint multivariate density. A crucial caveat with using the above estimates to do so is that it is not clear when the estimated node-conditional densities would be consistent with a joint multivariate density. There has been a line of work on this question (of when conditional densities are consistent with a joint density) for parametric densities; see [1] for an overview, with more recent results in [27, 5, 2, 25]. Overall, while estimating node-conditional densities could be viewed as surrogate estimation of a joint density, arbitrary node-conditional distributions need not be consistent in general with any joint density. There has however been a line of work in recent years [3, 28], where it was shown that when the node-conditional distributions belong to an exponential family, then under certain conditions on their parameterization, there do exist multivariate densities consistent with the node-conditional densities. In the next section, we leverage these results towards non-parametric estimation of conditional densities.

## 3  Conditional Densities of an Exponential Family Form

We first recall the definition of an *exponential family* in the context of a conditional density.

**Definition 1.** *A conditional density of a random variable $Y \in \mathcal{Y}$ given covariates $Z := (Z_1, \ldots, Z_m) \in \mathcal{Z}$ is said to have an exponential family form if it can be written as $f(Y \mid Z) = \exp(B(Y)^T E(Z) + C(Y) + D(Z))$, for some functions $B : \mathcal{Y} \mapsto \mathbb{R}^k$ (for some finite integer $k > 0$), $E : \mathcal{Z} \mapsto \mathbb{R}^k$, $C : \mathcal{Y} \mapsto \mathbb{R}$ and $D : \mathcal{Z} \mapsto \mathbb{R}$.*

Thus, $f(Y \mid Z)$ belongs to a finite-dimensional exponential family with sufficient statistics $B(Y)$, base measure $\exp(C(Y))$, and with natural parameter $E(Z)$ and where $-D(Z)$ is the log-partition function. Contrast this with a general conditional density $f(Y \mid Z) = \exp(h(Y, Z) + C(Y) + D(Z))$ with respect to the base measure $\exp(C(Y))$ and $-D(Z)$ being the log-normalization constant, and it can be seen that a conditional density of the exponential family form has its logistic density transform $h(Y, Z)$ that factorizes as $B(Y)^T E(Z)$.

Consider the case where the sufficient statistic function is real-valued. The non-parametric estimation problem of a conditional density of exponential form then reduces to the estimation of the sufficient statistics function $B(\cdot)$, the exponential natural parameter function $E(\cdot)$, assuming the base measure $C(\cdot)$ is given. But when would such estimated conditional densities be consistent with a joint density?

To answer this question, we draw upon developments in [28]. Suppose that the node-conditional distributions of each random variable $X_s$ conditioned on the rest of random variables have the exponential family form as in Definition 1, so that for each $s \in V$

$$\mathbb{P}(X_s \mid X_{-s}) \propto \exp\{E_s(X_{-s})B_s(X_s) + C_s(X_s)\}, \qquad (1)$$

for some arbitrary functions $E_s(\cdot), B_s(\cdot), C_s(\cdot)$ that specify a valid conditional density. Then [28] show that these node-conditional densities are consistent with a unique joint density over the random vector $X$, that moreover factors according to a set of cliques $\mathcal{C}$ in the graph $G$, **if and only if** the functions $\{E_s(\cdot)\}_{s \in V}$ specifying the node-conditional distributions have the form $E_s(X_{-s}) = \theta_s + \sum_{C \in \mathcal{C}: s \in C} \theta_C \prod_{t \in C, t \neq s} B_t(X_t)$, where $\{\theta_s\} \cup \{\theta_C\}_{C \in \mathcal{C}}$ is a set of parameters. Moreover, the corresponding consistent joint distribution has the following form

$$\mathbb{P}(X) \propto \exp\left\{\sum_{s \in V} \theta_s B_s(X_s) + \sum_{C \in \mathcal{C}} \theta_C \prod_{s \in C} B_s(X_s) + \sum_{s \in V} C_s(X_s)\right\}. \qquad (2)$$

In this paper, we are interested in the non-parametric estimation of the Expxorcist class of densities in (2), where we estimate both the finite-dimensional parameters $\{\theta_s\} \cup \{\theta_C\}_{C \in \mathcal{C}}$, as well as the functions $\{B_s(X_s)\}_{s \in V}$. We assume we are given the base measures $\{C_s(X_s)\}_{s \in V}$, so that the joint density is with respect to a given product base measure $\prod_{s \in V} \exp(C_s(X_S))$, as is common in the multivariate density estimation literature. Note that this is not a very restrictive assumption. In practice the base measure at each node can be well approximated using the empirical univariate marginal density of that node. We could also extend our algorithm, which we present next, to estimate the base measures along with sufficient statistic functions.

## 4 Regularized Conditional Likelihood Estimation for Exponential Family Form Densities

We consider the nonparametric estimation problem of estimating a joint density of the form in (2), focusing on the pairwise case where the factors have size at most $k = 2$, so that the joint density takes the form

$$\mathbb{P}(X) \propto \exp\left\{\sum_{s \in V} \theta_s B_s(X_s) + \sum_{(s,t) \in E} \theta_{st} B_s(X_s) B_t(X_t) + \sum_{s \in V} C_s(X_s)\right\}. \qquad (3)$$

As detailed in the previous section, estimating this joint density can be reduced to estimating its node-conditional densities, which take the form

$$\mathbb{P}(X_s \mid X_{-s}) \propto \exp\left\{B_s(X_s)\left(\theta_s + \sum_{t \in N_G(s)} \theta_{st} B_t(X_t)\right) + C_s(X_s)\right\}. \qquad (4)$$

We now introduce some notation which we use in the sequel. Let $\Theta = \{\theta_s\}_{s \in V} \cup \{\theta_{st}\}_{s \neq t}$ and $\Theta_s = \theta_s \cup \{\theta_{st}\}_{t \in V \setminus \{s\}}$. Let $B = \{B_s\}_{s \in V}$ be the set of sufficient statistics. Let $\mathcal{X}_s$ be the domain of $X_s$, which we assume is bounded and $L^2(\mathcal{X}_s)$ be the Hilbert space of square integrable functions over $\mathcal{X}_s$ with respect to Lebesgue measure. We assume that the sufficient statistics $B_s(\cdot) \in L^2(\mathcal{X}_s)$.

Note that the model in Equation (3) is unidentifiable. To overcome this issue we impose additional constraints on its parameters. Specifically, we require $B_s(X_s)$ to satisfy $\int_{\mathcal{X}_s} B_s(X)dX = 0$, $\int_{\mathcal{X}_s} B_s(X)^2 dX = 1$ and $\theta_s \geq 0, \forall s \in V$.

**Optimization objective:** Let $\mathbb{X}_n = \{X^{(1)}, \ldots X^{(n)}\}$ be $n$ i.i.d. samples drawn from a joint density of the form in Equation (3), with parameters $\Theta^*$, $B^*$. And let $\mathcal{L}_s(\Theta_s, B; \mathbb{X}_n)$ be the node conditional negative log likelihood at node $s$

$$\mathcal{L}_s(\Theta_s, B; \mathbb{X}_n) = \frac{1}{n}\sum_{i=1}^{n}\left\{-B_s(X_s^{(i)})\left(\theta_s + \sum_{t \in V \setminus s} \theta_{st} B_t(X_t^{(i)})\right) + A(X_{-s}^{(i)}; \Theta_s, B)\right\},$$

where $A(X_{-s}; \Theta_s, B)$ is the log partition function. To estimate the unknown parameters, we solve the following regularized node conditional log-likelihood estimation problem at each node $s \in V$

$$\min_{\Theta_s, B} \mathcal{L}_s(\Theta_s, B; \mathbb{X}_n) + \lambda_n \|\Theta_s\|_1$$
$$\text{s.t. } \theta_s \geq 0, \int_{\mathcal{X}_t} B_t(X)dX = 0, \int_{\mathcal{X}_t} B_t(X)^2 dX = 1 \quad \forall t \in V. \qquad (5)$$

The equality constraints on the norm of functions $B_t(\cdot)$ makes the above optimization problem a difficult one to solve. While the norm constraints on $B_t(\cdot), \forall t \in V \setminus s$ can be handled through re-parametrization, the constraint on $B_s(\cdot)$ can not be handled efficiently. To make the optimization more amenable for numerical optimization techniques, we solve a closely related optimization problem. At each node $s \in V$, we consider the following re-parametrization of $B$: $B_s(X_s) \leftarrow \theta_s B_s(X_s)$, $B_t(X_t) \leftarrow (\theta_{st}/\theta_s)B_t(X_t), \forall t \in V \setminus \{s\}$. With a slight abuse of notation we redefine $\mathcal{L}_s$ using this re-parametrization as

$$\mathcal{L}_s(B; \mathbb{X}_n) = \frac{1}{n} \sum\nolimits_{i=1}^{n} \left\{ -B_s(X_s^{(i)}) \left( 1 + \sum\nolimits_{t \in V \setminus s} B_t(X_t^{(i)}) \right) + A(X_{-s}^{(i)}; B) \right\}, \quad (6)$$

where $A(X_{-s}; B)$ is the log partition function. We solve the following optimization problem, which is closely related to the original optimization in Equation (5)

$$\min_{B} \ \mathcal{L}_s(B; \mathbb{X}_n) + \lambda_n \sum\nolimits_{t \in V} \sqrt{\int_{\mathcal{X}_t} B_t(X)^2 dX} \tag{7}$$
$$\text{s.t. } \int_{\mathcal{X}_t} B_t(X)dX = 0 \quad \forall t \in V.$$

For more details on the relation between (5) and (7), please refer to Appendix.

**Algorithm:** We now present our algorithm for optimization of (7). In the sequel, for simplicity, we assume that the domains $\mathcal{X}_t$ of random variables $X_t$ are all the same and equal to $\mathcal{X}$. In order to estimate functions $B_t$, we expand them over a uniformly bounded, orthonormal basis $\{\phi_k(\cdot)\}_{k=0}^{\infty}$ of $L^2(\mathcal{X})$ with $\phi_0(\cdot) \propto 1$. Expansion of the functions $B_t(\cdot)$ over this basis yields

$$B_t(X) = \sum\nolimits_{k=1}^{m} \alpha_{t,k}\phi_k(X) + \rho_{t,m}(X) \quad \text{where} \quad \rho_{t,m}(X) = \alpha_{t,0}\phi_0(X) + \sum\nolimits_{k=m+1}^{\infty} \alpha_{t,k}\phi_k(X).$$

Note that the constraint $\int_{\mathcal{X}} B_t(X)dX = 0$ in Equation (7), translates to $\alpha_{t,0} = 0$. To convert the infinite dimensional optimization problem in (7) into a finite dimensional problem, we truncate the basis expansion to the top $m$ terms and approximate $B_t(\cdot)$ as $\sum_{k=1}^{m} \alpha_{t,k}\phi_k(\cdot)$. The optimization problem in Equation (7) can then be rewritten as

$$\min_{\alpha_{\mathbf{m}}} \ \mathcal{L}_{s,m}(\alpha_{\mathbf{m}}; \mathbb{X}_n) + \lambda_n \sum_{t \in V} \|\alpha_{\mathbf{t,m}}\|_2, \tag{8}$$

where $\alpha_{\mathbf{t,m}} = \{\alpha_{t,k}\}_{k=1}^{m}$, $\alpha_{\mathbf{m}} = \{\alpha_{\mathbf{t,m}}\}_{t \in V}$ and $\mathcal{L}_{s,m}$ is defined as

$$\mathcal{L}_{s,m}(\alpha_{\mathbf{m}}; \mathbb{X}_n) = \frac{1}{n} \sum_{i=1}^{n} \left\{ -\sum_{k=1}^{m} \alpha_{s,k}\phi_k(X_s^{(i)}) \left( 1 + \sum_{t \in V \setminus \{s\}} \sum_{l=1}^{m} \alpha_{t,l}\phi_l(X_t^{(i)}) \right) + A(X_{-s}^{(i)}; \alpha_{\mathbf{m}}) \right\}.$$

**Iterative minimization of** (8): Note that the objective in (8) is non-convex. In this work, we use a simple alternating minimization technique for its optimization. In this technique, we alternately minimize $\alpha_{\mathbf{s,m}}$, $\{\alpha_{\mathbf{t,m}}\}_{t \in V \setminus s}$ while fixing the other parameters. The resulting optimization problem in each of the alternating steps is convex. We use Proximal Gradient Descent to optimize these sub-problems. To compute the objective and its gradients, we need to numerically evaluate the one-dimensional integrals in the log partition function. To do this, we choose a uniform grid of points over the domain and use quadrature rules to approximate the integrals.

**Convergence:** Although (8) is non-convex, we can show that under certain conditions on the objective function, the alternating minimization procedure converges to the global minimum. In a recent work [32] analyze alternating minimization for low rank matrix factorization problems and show that it converges to a global minimum if the sequence of convex problems are strongly convex and satisfy certain other regularity condition. The analysis of [32] can be extended to show global convergence of alternating minimization for (8).

## 5 Statistical Properties

In this section we provide parameter estimation error rates for the node conditional estimator in Equation (8). Note that these rates are for the re-parameterized model described in Equation (6) and can be easily translated to guarantees on the original model described in Equations (3), (4).

**Notation:** Let $\mathbb{B}_2(x, r) = \{y : \|y - x\|_2 \leq r\}$ be the $\ell_2$ ball with center $x$ and radius $r$. Let $\{B_t^*(\cdot)\}_{t \in V}$ be the true functions of the re-parametrized model, which we would like to estimate from the data. Denote the basis expansion coefficients of $B_t(\cdot)$ with respect to orthonormal basis $\{\phi_k(\cdot)\}_{k=0}^{\infty}$ by $\alpha_{\mathbf{t}}$, which is an infinite dimensional vector and let $\alpha_{\mathbf{t}}^*$ be the coefficients of $B_t^*(\cdot)$. And let $\alpha_{\mathbf{t},\mathbf{m}}$ be the coefficients corresponding to the top $m$ basis in the basis expansion of $B_t(\cdot)$. Note that $\int B_t(X)^2 dX = \|\alpha_{\mathbf{t}}\|_2^2$. Let $\alpha = \{\alpha_{\mathbf{t}}\}_{t \in V}$ and $\alpha_{\mathbf{m}} = \{\alpha_{\mathbf{t},\mathbf{m}}\}_{t \in V}$. Let $\bar{\mathcal{L}}_{s,m}(\alpha_{\mathbf{m}}) = \mathbb{E}\left[\mathcal{L}_{s,m}(\alpha_{\mathbf{m}}; \mathbb{X}_n)\right]$ be the population version of the sample loss defined in Equation (8). We will often omit $\mathbb{X}_n$ from $\mathcal{L}_{s,m}(\alpha_{\mathbf{m}}; \mathbb{X}_n)$ when clear from the context. We let $(\alpha_{\mathbf{t}} - \alpha_{\mathbf{t},\mathbf{m}})$ be the difference between infinite dimensional vector $\alpha_{\mathbf{t}}$ and the vector obtained by appropriately padding $\alpha_{\mathbf{t},\mathbf{m}}$ with zeros. Finally, we define the norm $\mathcal{R}(\cdot)$ as $\mathcal{R}(\alpha_{\mathbf{m}}) = \sum_{t \in V} \|\alpha_{\mathbf{t},\mathbf{m}}\|_2$ and its dual as $\mathcal{R}^*(\alpha_{\mathbf{m}}) = \sup_{t \in V} \|\alpha_{\mathbf{t},\mathbf{m}}\|_2$. The norms on infinite dimensional vector $\alpha$ are similarly defined.

We now state our key assumption on the loss function $\mathcal{L}_{s,m}(\cdot)$. This assumption imposes strong curvature condition on $\mathcal{L}_{s,m}$ along certain directions in a ball around $\alpha_{\mathbf{m}}^*$.

**Assumption 1.** *There exists $r_m > 0$ and constants $c, \kappa > 0$ such that for any $\Delta_{\mathbf{m}} \in \mathbb{B}_2(0, r_m)$ the gradient of the sample loss $\mathcal{L}_{s,m}$ satisfies:* $\langle \nabla \mathcal{L}_{s,m}(\alpha_{\mathbf{m}}^* + \Delta_{\mathbf{m}}) - \nabla \mathcal{L}_{s,m}(\alpha_{\mathbf{m}}^*), \Delta_{\mathbf{m}} \rangle \geq \kappa \|\Delta_{\mathbf{m}}\|_2^2 - c\sqrt{\frac{m \log(p)}{n}} \mathcal{R}(\Delta_{\mathbf{m}})$.

Similar assumptions are increasingly common in analysis of non-convex estimators, see [19] and references therein. We are now ready to state our results which give the parameter estimation error rates, the proofs of which can be found in Appendix. We first provide a deterministic bound on the error $\|\alpha_{\mathbf{m}} - \alpha_{\mathbf{m}}^*\|_2$ in terms of the random quantity $\mathcal{R}^*(\nabla \mathcal{L}_{s,m}(\alpha_{\mathbf{m}}^*))$. We derive probabilistic results in the subsequent corollaries.

**Theorem 2.** *Let $\mathcal{N}_s$ be the true neighborhood of node $s$, with $|\mathcal{N}_s| = d$. Suppose $\mathcal{L}_{s,m}$ satisfies Assumption 1. If the regularization parameter $\lambda_n$ is chosen such that $\lambda_n \geq 2\mathcal{R}^*(\nabla \mathcal{L}_{s,m}(\alpha_{\mathbf{m}}^*)) + 2c\sqrt{\frac{m \log(p)}{n}}$, then any stationary point $\hat{\alpha}_{\mathbf{m}}$ of (8) in $\mathbb{B}_2(\alpha_{\mathbf{m}}^*, r_m)$ satisfies:*

$$\|\hat{\alpha}_{\mathbf{m}} - \alpha_{\mathbf{m}}^*\|_2 \leq \frac{6\sqrt{2}}{\kappa} \sqrt{d} \lambda_n.$$

We now provide a set of sufficient conditions under which the random quantity $\mathcal{R}^*(\nabla \mathcal{L}_{s,m}(\alpha_{\mathbf{m}}^*))$ can be bounded.

**Assumption 2.** *There exists a constant $L > 0$ such that the gradient of the population loss $\bar{\mathcal{L}}_{s,m}$ at $\alpha_{\mathbf{m}}^*$ satisfies:* $\mathcal{R}^*(\nabla \bar{\mathcal{L}}_{s,m}(\alpha_{\mathbf{m}}^*)) \leq L\mathcal{R}^*(\alpha^* - \alpha_{\mathbf{m}}^*)$.

**Corollary 3.** *Suppose the conditions in Theorem 2 are satisfied. Moreover, let $\gamma = \sup_{i \in \mathbb{N}, X \in \mathcal{X}} |\phi_i(X)|$ and $\tau_m = \sup_{t \in V, X \in \mathcal{X}} |\sum_{i=1}^{m} \alpha_{t,i}^* \phi_i(X)|$. Suppose $\mathcal{L}_{s,m}$ satisfies Assumption 2. If the regularization parameter $\lambda_n$ is chosen such that $\lambda_n \geq 2L\mathcal{R}^*(\alpha^* - \alpha_{\mathbf{m}}^*) + c\gamma \tau_m \sqrt{\frac{md^2 \log(p)}{n}}$, then then with probability at least $1 - 2m/p^2$ any stationary point $\hat{\alpha}_{\mathbf{m}}$ of (8) in $\mathbb{B}_2(\alpha_{\mathbf{m}}^*, r_m)$ satisfies:*

$$\|\hat{\alpha}_{\mathbf{m}} - \alpha_{\mathbf{m}}^*\|_2 \leq \frac{6\sqrt{2}}{\kappa} \sqrt{d} \lambda_n.$$

Theorem 2 and Corollary 3 bound the error of the estimated coefficients in the truncated expansion. The approximation error of the truncated expansion itself depends on the function space assumption, as well as the basis chosen, but can be simply combined with the statement of the above corollary to derive the overall error. As an instance, we present a corollary below for the specific case of Sobolev space of order two, and the trigonometric basis.

**Corollary 4.** *Suppose the conditions in Corollary 3 are satisfied. Moreover, suppose the true functions $B_t^*(\cdot)$ lie in a Sobolev space of order two. Let $\{\phi_k\}_{k=0}^{\infty}$ be the trigonometric basis of $L^2(\mathcal{X})$. If the optimization problem (8) is solved with $\lambda_n = c_1(d^2 \log(p)/n)^{2/5}$ and $m = c_2(n/d^2 \log(p))^{1/5}$, then with probability at least $1 - 2m/p^2$ any stationary point $\hat{\alpha}_{\mathbf{m}}$ of (8) in $\mathbb{B}_2(\alpha_{\mathbf{m}}^*, r_m)$ satisfies:*

$$\|\hat{\alpha}_{\mathbf{m}} - \alpha^*\|_2 \leq c_3 \left(\frac{d^{13/4} \log(p)}{n}\right)^{2/5},$$

*where $c_1, c_2, c_3$ depend on $L, \kappa, \gamma, \tau_m$.*

**Discussion on Assumption 1:**   We now provide a set of sufficient conditions which ensure the restricted strong convexity (RSC) condition. Suppose the population risk $\bar{\mathcal{L}}_{s,m}(\cdot)$ is strongly convex in a ball of radius $r_m$ around $\alpha_{\mathbf{m}}^*$

$$\left\langle \nabla\bar{\mathcal{L}}_{s,m}(\alpha_{\mathbf{m}}^* + \Delta_{\mathbf{m}}) - \nabla\bar{\mathcal{L}}_{s,m}(\alpha_{\mathbf{m}}^*), \Delta_{\mathbf{m}} \right\rangle \geq \kappa\|\Delta_{\mathbf{m}}\|_2^2 \quad \forall \Delta_m \in \mathbb{B}_2(0, r_m). \tag{9}$$

Moreover, suppose the empirical gradients converge uniformly to the population gradients

$$\sup_{\alpha_{\mathbf{m}} \in \mathbb{B}_2(\alpha_{\mathbf{m}}^*, r_m)} \mathcal{R}^* \left( \nabla\mathcal{L}_{s,m}(\alpha_{\mathbf{m}}) - \nabla\bar{\mathcal{L}}_{s,m}(\alpha_{\mathbf{m}}) \right) \leq c\sqrt{\frac{m \log p}{n}}. \tag{10}$$

For example, this condition holds with high probability when the gradient of $\mathcal{L}_{s,m}(\alpha_{\mathbf{m}})$ w.r.t $\alpha_{\mathbf{t},\mathbf{m}}$, for any $t \in [p]$ is a sub-Gaussian process. Equations (9),(10) are easier to check and ensure that $\mathcal{L}_{s,m}(\alpha_{\mathbf{m}})$ satisfies the RSC property in Assumption 1.

# 6   Connections to Exponential Family MRF Copulas

The Expxorcist class of models could be viewed as being closely related to an exponential family MRF [28] copula density. Consider the parametric exponential family MRF joint density in (3): $\mathbb{P}_{\text{MRF};\theta}(X) \propto \exp\left\{ \sum_{s \in V} \theta_s B_s(X_s) + \sum_{(s,t) \in E(G)} \theta_{st} B_s(X_s) B_t(X_t) + \sum_{s \in V} C_s(X_s) \right\}$, where the distribution is indexed by the finite-dimensional parameters $\{\theta_s\}_{s \in V}, \{\theta_{st}\}_{(s,t) \in E}$, and where in contrast to the previous sections, we assume we are given the sufficient statistics functions $\{B_s(\cdot)\}_{s \in V}$ as well as the nodewise base measures $\{C_s(\cdot)\}_{s \in V}$. Now consider the following non-parametric problem. Given a random vector $X$, suppose we are interested in estimating monotonic node-wise functions $\{f_s(X_s)\}_{s \in V}$ such that $(f_1(X_1), \ldots, f_p(X_p))$ follows $\mathbb{P}_{\text{MRF};\theta}$ for some $\theta$. Letting $\mathbf{f}(X) = (f_1(X_1), \ldots, f_p(X_p))$, we have that $\mathbb{P}(\mathbf{f}(X)) = \mathbb{P}_{\text{MRF};\theta}(\mathbf{f}(X))$, so that the density of $X$ can be written as $\mathbb{P}(X) \propto \mathbb{P}(\mathbf{f}(X)) \prod_{s \in V} f_s'(X_s)$. This is now a semi-parametric estimation problem, where the unknowns are the functions $\{f_s(X_s)\}_{s \in V}$ as well as the finite-dimensional parameters $\theta$. To simplify this density, suppose we assume that the given node-wise sufficient statistics are linear, so that $B_s(z) = z$, for all $s \in V$, so that density reduces to

$$\mathbb{P}(X) \propto \exp\left\{ \sum_{s \in V} \theta_s f_s(X_s) + \sum_{(s,t) \in E(G)} \theta_{st} f_s(X_s) f_t(X_t) + \sum_{s \in V} (C_s(f_s(X_s)) + \log f_s'(X_s)) \right\}. \tag{11}$$

In contrast, the Expxorcist nonparametric exponential family graphical model takes the form

$$\mathbb{P}(X) \propto \exp\left\{ \sum_{s \in V} \theta_s f_s(X_s) + \sum_{(s,t) \in E(G)} \theta_{st} f_s(X_s) f_t(X_t) + \sum_{s \in V} C_s(X_s) \right\}. \tag{12}$$

It can be seen that the two densities have very similar forms, except that the density in (11) has a more complex base measure that depends on the unknown functions $\{f_s\}_{s \in V}$ and importantly the functions $\{f_s\}_{s \in V}$ in (11) are monotonic.

The class of densities in (11) can be cast as an exponential family MRF copula density. Suppose we denote the CDF of the parametric exponential family MRF joint density by $F_{\text{MRF};\theta}(X)$, with nodewise marginal CDFs $F_{\text{MRF};\theta,s}(X_s)$. Then the marginal CDF of the density (11) can be written as $F_s(x_s) = \mathbb{P}[X_s \leq x_s] = \mathbb{P}[f_s(X_s) \leq f_s(x_s)] = F_{\text{MRF};\theta,s}(f_s(x_s))$, so that

$$f_s(x_s) = F_{\text{MRF};\theta,s}^{-1}(F_s(x_s)). \tag{13}$$

It then follows that: $F(X) = F_{\text{MRF};\theta}\left( F_{\text{MRF};\theta,1}^{-1}(F_1(X_1)), \ldots, F_{\text{MRF};\theta,p}^{-1}(F_p(X_p)) \right)$, where $F(X)$ is the CDF of density (11). By letting $F_{\text{COP};\theta}(U) = F_{\text{MRF};\theta}\left( F_{\text{MRF};\theta,1}^{-1}(U_1), \ldots, F_{\text{MRF};\theta,p}^{-1}(U_p) \right)$ be the exponential family MRF copula density function, we see that the CDF of $X$ is precisely: $F(X) = F_{\text{COP};\theta}(F_1(X_1), \ldots, F_p(X_p))$, which is specified by the marginal CDFs $\{F_s(X_s)\}_{s \in V}$ and the copula density $F_{\text{COP};\theta}$ corresponding to the exponential family MRF density. In other words, the non-parametric extension in (11) of the exponential family MRF densities is precisely an exponential family MRF copula density. This development thus generalizes the non-parametric extension of Gaussian MRF densities via the Gaussian copula nonparanormal densities [17]. The caveats with the copula density however are two-fold: the node-wise functions are restricted to be monotonic, but

also the estimation of these as in (13) requires the estimation of inverses of marginal CDFs of an exponential family MRF, which is intractable in general. Thus, minor differences in the expressions of the Expxorcist density (12) and an exponential family MRF copula density (11) nonetheless have seemingly large consequences for tractable estimation of these densities from data.

# 7 Experiments

We present experimental results on both synthetic and real datasets. We compare our estimator, Expxorcist, with the Nonparanormal model of [17] and Gaussian Graphical Model (GGM). We use glasso [7] to estimate GGM and the two step estimator of [17] to estimate Nonparanormal model.

## 7.1 Synthetic Experiments

**Data:** We generated synthetic data from the Expxorcist model with chain and grid graph structures. For both the graph structures, we set $\theta_s = 1, \forall s \in V, \theta_{st} = 1, \forall (s,t) \in E$ and fix the domain $\mathcal{X}$ to $[-1,1]$. We experimented with two choices for sufficient statistics $B_s(X)$: $\sin(4\pi X)$ and $\left[\exp\left(-20(X-0.5)^2\right) + \exp\left(-20(X+0.5)^2\right) - 1\right]$ and picked the log base measure $C_s(X)$ to be 0. The grid graph we considered has a $10 \times (p/10)$ structure. We used Gibbs sampling to sample data from these models. We also generated data from Gaussian distribution with chain and grid graph structures. To generate this data we set the off diagonal non-zero entries of inverse covariance matrix to 0.49 for chain graph and 0.25 for grid graph and diagonal entries to 1.

**Evaluation Metric:** We compared the performance of Expxorcist against baselines, on graph structure recovery, using ROC curves. The ROC curve plots the *true positive rate* (TPR) against *false positive rate* (FPR) over different choices of regularization parameter, where TPR is the fraction of correctly detected edges and FPR is the fraction of mis-identified non edges.

**Experiment Settings:** For this experiment we set $p = 50$ and $n \in \{100, 200, 500\}$ and varied the regularization parameter $\lambda$ from $10^{-2}$ to 1. To fit the data to the non parametric model (3), we used cosine basis and truncated the basis expansion to top 30 terms. In practice, one could choose the number of basis ($m$) based on domain knowledge (e.g. "smooth" functions), or in the absence of which, one could use hold-out validation/cross validation. Given $\hat{N}(s)$, the estimated neighborhood for node $s$, we estimated the overall graph structure as: $\cup_{s \in V} \cup_{t \in \hat{N}(s)} \{(s,t)\}$. To reduce the variance in the ROC plots, we averaged results over 10 repetitions.

**Results:** Figure 1 shows the ROC plots obtained from this experiment. Due to the lack of space, we present more experimental results in Appendix. It can be seen that Expxorcist has much better performance on non-Gaussian data. On these datasets, even at $n = 500$ the baselines chose edges at random. This suggests that in the presence of multiple modes and fat tails, Expxorcist is a better model. Expxorcist has slightly poor performance than baselines on Gaussian data. However, this is expected because it learns a broader family of distributions than Nonparanormal.

## 7.2 Futures Intraday Data

We now present our analysis on the Futures price returns. This dataset was downloaded from `http://www.kibot.com/`. We focus on the Top-26 most liquid instruments being traded at the Chicago Mercantile Exchange (CME). The instruments span different sectors like Energy, Agriculture, Currencies, Equity Indices, Metals and Interest Rates. We focus on the hours of maximum liquidity (9am Eastern to 3pm Eastern) and look at the 1 minute price returns. The return distribution is a mixture of 1 minute returns with the overnight return. Since overnight returns tend to be bigger than the 1 minute return within the day, the return distribution is multimodal and fat-tailed. We treat each instrument as a random variable and the 1 minute returns as independent samples drawn from these random variables. We use the data collected in February 2010 as training data and data from March 2010 as held out data for tuning parameter selection. After removing samples with missing entries we are left with 894 training and 650 held out data samples. We fit Expxorcist and baselines on this data with the same parameter settings described above. For each of these models, we select the best tuning parameter through log likelihood on held out data. However, this criteria resulted in complete graphs for Nonparanormal and GGM (325 edges) and a relatively sparser graph for Expxorcist (168 edges). So for a better comparison of these models, we selected tuning parameters for each of the models such that the resulting graphs have almost the same number of edges. Figure 2 shows the

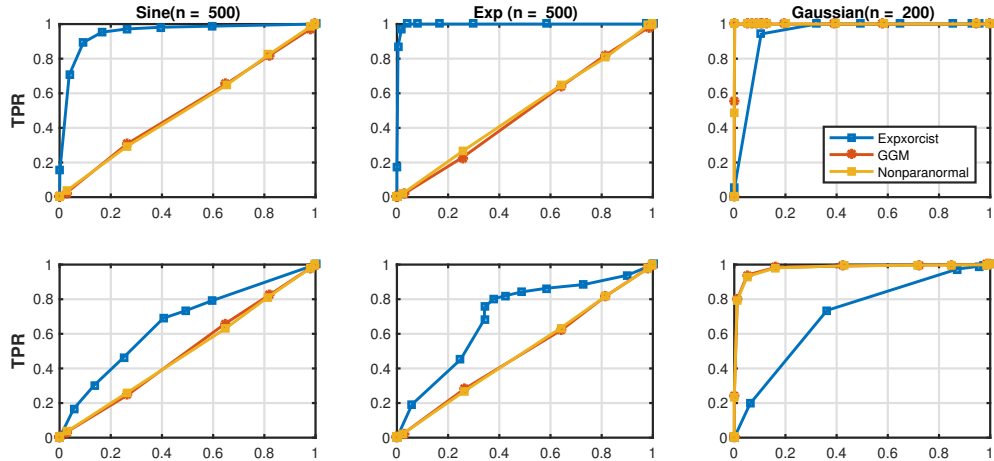

Figure 1: ROC plots from synthetic experiments. Top and bottom rows show plots for chain and grid graphs respectively. Left column shows plots for data generated from our non-parametric model with $B_s(X) = \sin(X)$, $n = 500$ and center column shows plots for the other choice of sufficient statistic with $n = 500$. Right column shows plots for Gaussian data with $n = 200$.

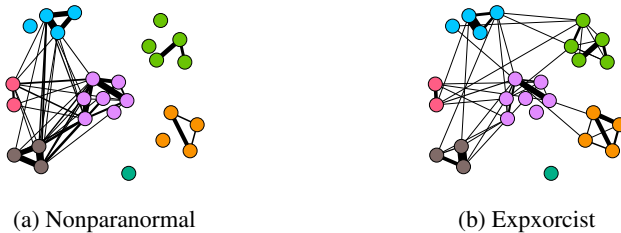

(a) Nonparanormal        (b) Expxorcist

Figure 2: Graph Structures learned for the Futures Intraday Data. The Expxorcist graph shown here was obtained by selecting $\lambda = 0.1$. Nodes are colored based on their categories. Edge thickness is proportional to the magnitude of the interaction.

learned graphs for one such choice of tuning parameters, which resulted in $\sim 52$ edges in the graphs. Nonparanormal and GGM resulted in very similar graphs, so we only present Nonparanormal here. It can be seen that Expxorcist is able to identify the clusters better than Nonparanormal. More detailed graphs and comparison with GGM can be found in Appendix.

# 8 Conclusion

In this work we considered the problem of non-parametric density estimation and introduced Expxorcist, a new family of non-parametric graphical models. Our approach relies on a simple function space assumption that the conditional distribution of each variable conditioned on the other variables has a non-parametric exponential family form. We proposed an estimator for Expxorcist that is computationally efficient and comes with statistical guarantees. Our empirical results suggest that, in the presence of multiple modes and fat tails in the data, our non-parametric model is a better choice than the Nonparanormal model of [17].

# 9 Acknowledgement

A.S. and P.R. acknowledge the support of ARO via W911NF-12-1-0390 and NSF via IIS-1149803, IIS-1447574, DMS-1264033, and NIH via R01 GM117594-01 as part of the Joint DMS/NIGMS Initiative to Support Research at the Interface of the Biological and Mathematical Sciences. M. K. acknowledges support by an IBM Corporation Faculty Research Fund at the University of Chicago Booth School of Business.

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
