[Supplementary Material]

# Supplementary material for Nonparametric Graphical Models Via Conditional Exponential Densities

**Arun Sai Suggala**
Carnegie Mellon University
Pittsburgh, PA 15213
asuggala@cs.cmu.edu

**Mladen Kolar**
University of Chicago
Chicago, IL 60637
mkolar@chicagobooth.edu

**Pradeep Ravikumar**
Carnegie Mellon University
Pittsburgh, PA 15213
pradeepr@cs.cmu.edu

## A  Node Conditional Maximum Likelihood Estimation

In this section we derive the relation between optimization problems (5) and (7) defined in Section 4 of the main paper. We start with the optimization problem (5), which is defined in terms of the original parameters of non-parametric graphical model (3):

$$\min_{\Theta_s, B} \mathcal{L}_s(\Theta_s, B; \mathbb{X}_n) + \lambda_n \|\Theta_s\|_1$$

$$\text{s.t. } \theta_s \geq 0, \int_{\mathcal{X}_t} B_t(X) dX = 0, \int_{\mathcal{X}_t} B_t(X)^2 dX = 1 \quad \forall t \in V,$$

where $\mathcal{L}_s(\Theta_s, B; \mathbb{X}_n)$ is the node conditional negative log likelihood at node $s$:

$$\mathcal{L}_s(\Theta_s, B; \mathbb{X}_n) = \frac{1}{n} \sum_{i=1}^n \left\{ -B_s(X_s^{(i)}) \left( \theta_s + \sum_{t \in V \backslash s} \theta_{st} B_t(X_t^{(i)}) \right) + A(X_{-s}^{(i)}; \theta_s, B) \right\}.$$

Let $\widetilde{B}_t(X_t) = \theta_{st} B_t(X_t), \forall t \in V \backslash \{s\}$. Using this parametrization $\mathcal{L}_s(\cdot)$ can be written as:

$$\mathcal{L}_s(\theta_s, B_s, \widetilde{B}_{-s}; \mathbb{X}_n) = \frac{1}{n} \sum_{i=1}^n \left\{ -B_s(X_s^{(i)}) \left( \theta_s + \sum_{t \in V \backslash s} \widetilde{B}_t(X_t^{(i)}) \right) + A(X_{-s}^{(i)}; \Theta_s, B_s, \widetilde{B}_{-s}) \right\}.$$

Note that, given $\widetilde{B}_t$, one can recover $\theta_{st}$ (and thus $B_t$) by computing the $L^2(\mathcal{X}_t)$ norm of $\widetilde{B}_t$. Using this re-parametrization the original optimization in Equation (5) can be written as the following equivalent problem:

$$\min_{\theta_s, B_s, \widetilde{B}_{-s}} \mathcal{L}_s(\theta_s, B_s, \widetilde{B}_{-s}; \mathbb{X}_n) + \lambda_n \sum_{t \in V \backslash s} \sqrt{\int_{\mathcal{X}_t} \widetilde{B}_t(X)^2 dX} + \lambda_n |\theta_s|$$

$$\text{s.t. } \theta_s \geq 0, \int_{\mathcal{X}_s} B_s(X)^2 dX = 1, \int_{\mathcal{X}_t} \widetilde{B}_t(X) dX = 0 \quad \forall t \in V.$$

The above problem still has the equality constraint on the norm of functions $B_s(\cdot)$. As pointed out in the main paper, this makes the above optimization problem a difficult one to solve. To make

the optimization more amenable for numerical optimization techniques, we solve a closely related optimization problem. At each node $s \in V$, we consider the following re-parametrization of $B$: $B_s(X_s) \leftarrow \theta_s B_s(X_s)$, $B_t(X_t) \leftarrow (\theta_{st}/\theta_s)B_t(X_t) \forall t \in V \setminus \{s\}$. With a slight abuse of notation we redefine $\mathcal{L}_s$ using this re-parametrization as:

$$\mathcal{L}_s(B; \mathbb{X}_n) = \frac{1}{n} \sum_{i=1}^{n} \left\{ -B_s(X_s^{(i)}) \left( 1 + \sum_{t \in V \setminus s} B_t(X_t^{(i)}) \right) + A(X_{-s}^{(i)}; B) \right\},$$

where $A(X_{-s}; B)$ is the log partition function. We solve the following optimization problem, which is closely related to the original optimization in Equation (5):

$$\min_B \mathcal{L}_s(B; \mathbb{X}_n) + \lambda_n \sum_{t \in V} \sqrt{\int_{\mathcal{X}_t} B_t(X)^2 dX}$$

$$\text{s.t.} \int_{\mathcal{X}_t} B_t(X) dX = 0 \quad \forall t \in V.$$

Note that $\theta_s^*$ need to be bounded away from 0, for this re-parametrized optimization problem to give consistent estimates of the true parameters.

## B  Statistical Properties

### B.1  Proof of Theorem 2

*Proof.* Define $\mathbb{C} = \{\alpha_{\mathbf{m}}^* + \Delta : \mathcal{R}(\Delta_{\mathcal{N}_s^c}) \leq 3\mathcal{R}(\Delta_{\mathcal{N}_s})\}$, where $\Delta_{\mathcal{N}_s}$ is the sub-vector of $\Delta$ restricted to the coordinates specified by variables $\{\alpha_{\mathbf{t},\mathbf{m}} : t \in \mathcal{N}_s \cup \{s\}\}$. Let $\mathcal{F}_{s,m}$ denote the optimization objective in Equation (8):

$$\mathcal{F}_{s,m}(\alpha_{\mathbf{m}}; \mathbb{X}_n) = \mathcal{L}_{s,m}(\alpha_{\mathbf{m}}; \mathbb{X}_n) + \lambda_n \mathcal{R}(\alpha_{\mathbf{m}}).$$

We prove the theorem in two parts. In the first part we show that $\mathcal{F}_{s,m}$ doesn't have any stationary points in $\mathbb{B}_2(\alpha_{\mathbf{m}}^*, r_m) \cap \mathbb{C}^c$. In the second part we show that $\mathcal{F}_{s,m}$ doesn't have any stationary points in $\mathbb{B}_2(\alpha_{\mathbf{m}}^*, r_m) \cap \mathbb{C} \setminus \mathbb{B}_2(\alpha_{\mathbf{m}}^*, r_s)$, where $r_s = (6\sqrt{2}/\kappa)\sqrt{d}\lambda_n$. The proof of the Theorem then follows by combining the results from these two parts.

**(a) No stationary points in $\mathbb{B}_2(\alpha_{\mathbf{m}}^*, r_m) \cap \mathbb{C}^c$:**  Let $\alpha_{\mathbf{m}} \in \mathbb{B}_2(\alpha_{\mathbf{m}}^*, r_m) \cap \mathbb{C}^c$ and let $\Delta = \alpha_{\mathbf{m}} - \alpha_{\mathbf{m}}^*$. Let $\partial f(x)$ denote the set of sub-gradients of a function $f(.)$ at $x$. For any $z(\alpha_{\mathbf{m}}) \in \partial \mathcal{F}_{s,m}(\alpha_{\mathbf{m}})$, where $z(\alpha_{\mathbf{m}}) = \nabla \mathcal{L}_{s,m}(\alpha_{\mathbf{m}}) + \lambda_n v(\alpha_{\mathbf{m}})$, $v(\alpha_{\mathbf{m}}) \in \partial \mathcal{R}(\alpha_{\mathbf{m}})$, we have:

$$
\begin{aligned}
\langle z(\alpha_{\mathbf{m}}), \alpha_{\mathbf{m}} - \alpha_{\mathbf{m}}^* \rangle &= \langle z(\alpha_{\mathbf{m}}), \Delta \rangle \\
&= \langle \nabla \mathcal{L}_{s,m}(\alpha_{\mathbf{m}}), \Delta \rangle + \lambda_n \langle v(\alpha_{\mathbf{m}}), \Delta \rangle \\
&= \langle \nabla \mathcal{L}_{s,m}(\alpha_{\mathbf{m}}) - \nabla \mathcal{L}_{s,m}(\alpha_{\mathbf{m}}^*), \Delta \rangle + \langle \nabla \mathcal{L}_{s,m}(\alpha_{\mathbf{m}}^*), \Delta \rangle + \lambda_n \langle v(\alpha_{\mathbf{m}}), \Delta \rangle,
\end{aligned}
\tag{1}
$$

We now bound each of the terms in the RHS of above equation. From Assumption 1 on RSC property of the sample loss, we have

$$\langle \nabla \mathcal{L}_{s,m}(\alpha_{\mathbf{m}}) - \nabla \mathcal{L}_{s,m}(\alpha_{\mathbf{m}}^*), \Delta \rangle \geq \kappa \|\Delta\|_2^2 - c\sqrt{\frac{m \log(p)}{n}} \mathcal{R}(\Delta).$$

From the definition of $\mathcal{N}_s$ we have $(\alpha_{\mathbf{m}}^*)_{\mathcal{N}_s^c} = 0$. So we have

$$\langle v(\alpha_{\mathbf{m}}), \Delta \rangle = \left\langle v(\alpha_{\mathbf{m}})_{\mathcal{N}_s^c}, (\alpha_{\mathbf{m}})_{\mathcal{N}_s^c} \right\rangle + \langle v(\alpha_{\mathbf{m}})_{\mathcal{N}_s}, \Delta_{\mathcal{N}_s} \rangle \geq \mathcal{R}(\Delta_{\mathcal{N}_s^c}) - \mathcal{R}(\Delta_{\mathcal{N}_s}),$$

where the last inequality follows from the properties of sub-gradient of the norm $\mathcal{R}(.)$. Finally, from the definition of dual norm $\mathcal{R}^*$, we have:

$$\langle \nabla \mathcal{L}_{s,m}(\alpha_{\mathbf{m}}^*), \Delta \rangle \geq -\mathcal{R}^*(\nabla \mathcal{L}_{s,m}(\alpha_{\mathbf{m}}^*))\mathcal{R}(\Delta).$$

Substituting these results in Equation (1) we get:

$$
\begin{aligned}
\langle z(\alpha_{\mathbf{m}}), \alpha_{\mathbf{m}} - \alpha_{\mathbf{m}}^* \rangle \quad \geq \quad & \kappa \|\Delta\|_2^2 - \mathcal{R}^*(\nabla \mathcal{L}_{s,m}(\alpha_{\mathbf{m}}^*)) \mathcal{R}(\Delta) \\
& - \left( c \sqrt{\tfrac{m \log(p)}{n}} \right) \mathcal{R}(\Delta) + \lambda_n \left( \mathcal{R}(\Delta_{\mathcal{N}_s^c}) - \mathcal{R}(\Delta_{\mathcal{N}_s}) \right).
\end{aligned}
\tag{2}
$$

Note that since $\alpha_{\mathbf{m}} \in \mathbb{C}^c$ we have $\mathcal{R}(\Delta_{\mathcal{N}_s^c}) \geq 3\mathcal{R}(\Delta_{\mathcal{N}_s})$. Moreover, from our choice of $\lambda_n$ we know that $\lambda_n \geq 2\mathcal{R}^*(\nabla \mathcal{L}_{s,m}(\alpha_{\mathbf{m}}^*)) + 2c\sqrt{\tfrac{m \log(p)}{n}}$. Substituting this in the above equation we get:

$$
\langle z(\alpha_{\mathbf{m}}), \alpha_{\mathbf{m}} - \alpha_{\mathbf{m}}^* \rangle \quad \geq \quad \kappa \|\Delta\|_2^2 + \left( \mathcal{R}(\Delta_{\mathcal{N}_s^c}) - 3\mathcal{R}(\Delta_{\mathcal{N}_s}) \right) \left( \mathcal{R}^*(\nabla \mathcal{L}_{s,m}(\alpha_{\mathbf{m}}^*)) + c\sqrt{\tfrac{m \log(p)}{n}} \right).
\tag{3}
$$

The above inequality shows that $\langle z(\alpha_{\mathbf{m}}), \alpha_{\mathbf{m}} - \alpha_{\mathbf{m}}^* \rangle > 0, \forall \alpha_{\mathbf{m}} \in \mathbb{B}_2(\alpha_{\mathbf{m}}^*, r_m) \cap \mathbb{C}^c$.

Now suppose $\alpha_{\mathbf{m}} \in \mathbb{B}_2(\alpha_{\mathbf{m}}^*, r_m) \cap \mathbb{C}^c$ is a stationary point. Then from *first-order necessary conditions* we know that $\langle z(\alpha_{\mathbf{m}}), \alpha_{\mathbf{m}}^* - \alpha_{\mathbf{m}} \rangle \geq 0$. However, this contradicts the result we obtained in Equation (3). This shows that there are no stationary points in $\mathbb{B}_2(\alpha_{\mathbf{m}}^*, r_m) \cap \mathbb{C}^c$.

**(b) No stationary points in $\mathbb{B}_2(\alpha_{\mathbf{m}}^*, r_m) \cap \mathbb{C} \setminus \mathbb{B}_2(\alpha_{\mathbf{m}}^*, r_s)$:** The proof of this part follows along the lines of the previous part. Let $\alpha_{\mathbf{m}} \in \mathbb{B}_2(\alpha_{\mathbf{m}}^*, r_m) \cap \mathbb{C} \setminus \mathbb{B}_2(\alpha_{\mathbf{m}}^*, r_s)$. From Equations (1), (2) we know that:

$$
\begin{aligned}
\langle z(\alpha_{\mathbf{m}}), \alpha_{\mathbf{m}} - \alpha_{\mathbf{m}}^* \rangle \quad \geq \quad & \kappa \|\Delta\|_2^2 - \mathcal{R}^*(\nabla \mathcal{L}_{s,m}(\alpha_{\mathbf{m}}^*)) \mathcal{R}(\Delta) \\
& - \left( c \sqrt{\tfrac{m \log(p)}{n}} \right) \mathcal{R}(\Delta) + \lambda_n \langle v(\alpha_{\mathbf{m}}), \Delta \rangle.
\end{aligned}
\tag{4}
$$

Since $\alpha_{\mathbf{m}} \in \mathbb{C}$ we have $\mathcal{R}(\Delta) \leq 4\sqrt{d+1}\|\Delta\|_2 \leq 4\sqrt{2d}\|\Delta\|_2$. Substituting this in the above equation we get:

$$
\begin{aligned}
\langle z(\alpha_{\mathbf{m}}), \alpha_{\mathbf{m}} - \alpha_{\mathbf{m}}^* \rangle \quad \geq \quad & \kappa \|\Delta\|_2^2 - \mathcal{R}^*(\nabla \mathcal{L}_{s,m}(\alpha_{\mathbf{m}}^*)) \mathcal{R}(\Delta) - \left( c \sqrt{\tfrac{m \log(p)}{n}} \right) \mathcal{R}(\Delta) - \lambda_n \mathcal{R}(\Delta) \\[2mm]
\geq \quad & \kappa \|\Delta\|_2^2 - \mathcal{R}(\Delta) \left( \mathcal{R}^*(\nabla \mathcal{L}_{s,m}(\alpha_{\mathbf{m}}^*)) + c\sqrt{\tfrac{m \log(p)}{n}} + \lambda_n \right) \\[2mm]
\geq \quad & \kappa \|\Delta\|_2^2 - 4\sqrt{2d}\|\Delta\|_2 \left( \mathcal{R}^*(\mathcal{L}_{s,m}(\alpha_{\mathbf{m}}^*)) + c\sqrt{\tfrac{m \log(p)}{n}} + \lambda_n \right) \\[2mm]
\geq \quad & \left( \kappa \|\Delta\|_2 - 4\sqrt{2d} \left( \mathcal{R}^*(\nabla \mathcal{L}_{s,m}(\alpha_{\mathbf{m}}^*)) + c\sqrt{\tfrac{m \log(p)}{n}} + \lambda_n \right) \right) \|\Delta\|_2 \\[2mm]
\geq \quad & \left( \kappa \|\Delta\|_2 - 4\sqrt{2d} \left( \tfrac{3}{2}\lambda_n \right) \right) \|\Delta\|_2.
\end{aligned}
\tag{5}
$$

The above inequality shows that $\langle z(\alpha_{\mathbf{m}}), \alpha_{\mathbf{m}} - \alpha_{\mathbf{m}}^* \rangle > 0, \forall \alpha_{\mathbf{m}} \in \mathbb{B}_2(\alpha_{\mathbf{m}}^*, r_m) \cap \mathbb{C} \setminus \mathbb{B}_2(\alpha_{\mathbf{m}}^*, r_s)$. This shows that there are no stationary points in $\mathbb{B}_2(\alpha_{\mathbf{m}}^*, r_m) \cap \mathbb{C} \setminus \mathbb{B}_2(\alpha_{\mathbf{m}}^*, r_s)$.

Following results from parts (a) and (b) we conclude that any stationary point in $\mathbb{B}_2(\alpha_{\mathbf{m}}^*, r_m)$ satisfies $\|\alpha_{\mathbf{m}} - \alpha_{\mathbf{m}}^*\|_2 \leq \frac{6\sqrt{2}}{\kappa}\sqrt{d}\lambda_n$. $\qquad\square$

## B.2 Proof of Corollary 3

Before we proceed to the proof of Corollary 3, we introduce some notation we use in its proof. We say $Z$ is a $\sigma-$sub-Gaussian random variable, if it satisfies the following tail property:

$$
\mathbb{P}\left( |Z - \mathbb{E}[Z]| \geq \epsilon \right) \leq 2\exp\{-\frac{\epsilon^2}{2\sigma^2}\}.
$$

We use the notation $Z \in SG(\sigma^2)$ to say that a random variable $Z$ is $\sigma-$sub-Gaussian . We use the following standard result from concentration theory: if $Z_1 \ldots Z_n$ are $n$ i.i.d $SG(\sigma^2)$ random variables then $\frac{1}{n}\sum_{i=1}^n Z_i \in SG(\frac{\sigma^2}{n})$.

The following Lemma provides an upper bound for $\mathcal{R}^*(\nabla \mathcal{L}_{s,m}(\alpha_{\mathbf{m}}^*))$ that holds with high probability.

**Lemma 1.** *Let $\mathcal{N}_s$ be the true neighborhood of node $s$, with $|\mathcal{N}_s| = d$. Moreover, let*

$$\gamma = \sup_{i \in \mathbb{N}, X \in \mathcal{X}} |\phi_i(X)|$$

*and*

$$\tau_m = \sup_{t \in V, X \in \mathcal{X}} |\sum_{i=1}^{m} \alpha_{t,i}^* \phi_i(X)|.$$

*Suppose $\mathcal{L}_{s,m}$ satisfies Assumption 2. Then with probability at least $1 - 2m/p^2$ we have:*

$$\mathcal{R}^*(\nabla \mathcal{L}_{s,m}(\alpha_{\mathbf{m}}^*)) \leq L\mathcal{R}^*(\alpha^* - \alpha_{\mathbf{m}}^*) + c_1 \gamma \tau_m \sqrt{\frac{md^2 \log(p)}{n}},$$

*where $c_1 > 0$ is a constant.*

*Proof.* From triangle inequality, we have:

$$\mathcal{R}^*(\nabla \mathcal{L}_{s,m}(\alpha_{\mathbf{m}}^*)) \leq \mathcal{R}^*(\nabla \mathcal{L}_{s,m}(\alpha_{\mathbf{m}}^*) - \nabla \bar{\mathcal{L}}_{s,m}(\alpha_{\mathbf{m}}^*)) + \mathcal{R}^*(\nabla \bar{\mathcal{L}}_{s,m}(\alpha_{\mathbf{m}}^*)).$$

We now upper bound each of the terms in the RHS of the above equation.

**(a) Upper bound for $\mathcal{R}^*(\nabla \mathcal{L}_{s,m}(\alpha_{\mathbf{m}}^*) - \nabla \bar{\mathcal{L}}_{s,m}(\alpha_{\mathbf{m}}^*))$:** The gradient of $\mathcal{L}_{s,m}(\alpha_{\mathbf{m}})$ at $\alpha_{\mathbf{m}}^*$ is given by:

$$(\nabla \mathcal{L}_{s,m}(\alpha_{\mathbf{m}}^*))_{s,k} = \frac{1}{n} \sum_{i=1}^{n} -\left(\phi_k(X_s^{(i)})E_{-s}(X_{-s}^{(i)};\alpha_{\mathbf{m}}^*)\right) + \mathbb{E}_{\alpha_{\mathbf{m}}^*}\left[\phi_k(X_s)E_{-s}(X_{-s}^{(i)};\alpha_{\mathbf{m}}^*)\,\Big|\, X_{-s}^{(i)}\right]$$

and for $t \in V \setminus \{s\}$:

$$(\nabla \mathcal{L}_{s,m}(\alpha_{\mathbf{m}}^*))_{t,k} = \frac{1}{n} \sum_{i=1}^{n} -\left(\phi_k(X_t^{(i)})B_s(X_s^{(i)};\alpha_{\mathbf{m}}^*)\right) + \mathbb{E}_{\alpha_{\mathbf{m}}^*}\left[\phi_k(X_t^{(i)})B_s(X_s;\alpha_{\mathbf{m}}^*)\,\Big|\, X_{-s}^{(i)}\right],$$

where $E_{-s}(X_{-s};\alpha_{\mathbf{m}}) = 1 + \sum_{t \in V \setminus \{s\}} \sum_{j=1}^{m} \alpha_{t,j}\phi_j(X_t)$ and $B_s(X_s;\alpha_{\mathbf{m}}) = \sum_{j=1}^{m} \alpha_{s,j}\phi_j(X_s)$ and $\mathbb{E}_\alpha[.]$ denotes expectation with respect to the density parametrized by $\alpha$ and $(\nabla \mathcal{L}_{s,m}(\alpha_{\mathbf{m}}^*))_{t,k}$ is the gradient of $\mathcal{L}_{s,m}(\alpha_{\mathbf{m}})$ evaluated at $\alpha_{\mathbf{m}}^*$ with respect to variable $\alpha_{t,k}$.

We now show that $(\nabla \mathcal{L}_{s,m}(\alpha_{\mathbf{m}}^*))_{t,k}$ concentrates well around its expectation. Note that

$$\mathbb{E}\left[(\nabla \mathcal{L}_{s,m}(\alpha_{\mathbf{m}}^*))_{t,k}\right] = \left(\nabla \bar{\mathcal{L}}_{s,m}(\alpha_{\mathbf{m}}^*)\right)_{t,k}.$$

Lets first define random variables $\{Y_{s,k}\}_{k=1}^{m}$ and $\{Y_{t,k}\}_{k=1}^{m}, \forall t \in V \setminus \{s\}$ as:

$$Y_{s,k}(X^{(i)}) = -\left(\phi_k(X_s^{(i)})E_{-s}(X_{-s}^{(i)};\alpha_{\mathbf{m}}^*)\right) + \mathbb{E}_{\alpha_{\mathbf{m}}^*}\left[\phi_k(X_s)E_{-s}(X_{-s}^{(i)};\alpha_{\mathbf{m}}^*)\,\Big|\, X_{-s}^{(i)}\right],$$

and

$$Y_{t,k}(X^{(i)}) = -\left(\phi_k(X_t^{(i)})B_s(X_s^{(i)};\alpha_{\mathbf{m}}^*)\right) + \mathbb{E}_{\alpha_{\mathbf{m}}^*}\left[\phi_k(X_t^{(i)})B_s(X_s;\alpha_{\mathbf{m}}^*)\,\Big|\, X_{-s}^{(i)}\right].$$

To ease the notation, we denote $Y_{s,k}(X^{(i)})$ by $Y_{s,k}^{(i)}$. We now rewrite $\nabla \mathcal{L}_{s,m}(\alpha_{\mathbf{m}}^*)$ in terms of random variables $\{Y_{s,k}\}_{k=1}^{m}$ and $\{Y_{t,k}\}_{k=1}^{m}$ as follows:

$$(\nabla \mathcal{L}_{s,m}(\alpha_{\mathbf{m}}^*))_{s,k} = \frac{1}{n} \sum_{i=1}^{n} Y_{s,k}^{(i)}, \qquad (\nabla \mathcal{L}_{s,m}(\alpha_{\mathbf{m}}^*))_{t,k} = \frac{1}{n} \sum_{i=1}^{n} Y_{t,k}^{(i)}.$$

For any $k \in [1, m], i \in [1, n]$ we have:

$$\begin{aligned}
|Y_{s,k}^{(i)}| &\leq \left|\left(\phi_k(X_s^{(i)})E_{-s}(X_{-s}^{(i)};\alpha_{\mathbf{m}}^*)\right)\right| + \left|\mathbb{E}_{\alpha_{\mathbf{m}}^*}\left[\phi_k(X_s)E_{-s}(X_{-s}^{(i)};\alpha_{\mathbf{m}}^*)\,\Big|\, X_{-s}^{(i)}\right]\right| \\
&\leq \gamma\left|\left(E_{-s}(X_{-s}^{(i)};\alpha_{\mathbf{m}}^*)\right)\right| + \gamma\mathbb{E}_{\alpha_{\mathbf{m}}^*}\left[\left|E_{-s}(X_{-s}^{(i)};\alpha_{\mathbf{m}}^*)\right|\,\Big|\, X_{-s}^{(i)}\right],
\end{aligned}$$

where the last inequality follows from Jensen's inequality and the fact that $\gamma = \sup\limits_{j\in\mathbb{N}, X\in\mathcal{X}} |\phi_j(X)|$.

We now bound $\left|\left(E_{-s}(X^{(i)}_{-s}; \alpha^*_\mathbf{m})\right)\right|$:

$$
\begin{aligned}
\left|\left(E_{-s}(X^{(i)}_{-s}; \alpha^*_\mathbf{m})\right)\right| &= \left|1 + \sum_{t\in V\setminus\{s\}}\sum_{j=1}^{m} \alpha^*_{t,j}\phi_j(X^{(i)}_t)\right| \\
&\leq 1 + \sum_{t\in V\setminus\{s\}}\left|\sum_{j=1}^{m} \alpha^*_{t,j}\phi_j(X^{(i)}_t)\right| \\
&\leq 1 + d\tau_m \leq d(1 + \tau_m).
\end{aligned}
$$

Substituting this in the above equation we get:

$$
|Y^{(i)}_{s,k}| \leq 2\gamma d(1 + \tau_m).
$$

Using similar arguments we can show that $|Y^{(i)}_{t,k}| \leq 2\gamma\tau_m, \forall t\in V\setminus\{s\}$. This shows that $Y^{(i)}_{s,k} \in$ SG$(4\gamma^2 d^2(1 + \tau_m)^2)$ and $Y^{(i)}_{t,k} \in$ SG$(4\gamma^2\tau_m^2), \forall t\in V\setminus\{s\}$. Since $(\nabla\mathcal{L}_{s,m}(\alpha^*_\mathbf{m}))_{t,k}$ is a sum of $n$ i.i.d random variables $\{Y^{(i)}_{t,k}\}_{i=1}^{n}$ we have:

$$
(\nabla\mathcal{L}_{s,m}(\alpha^*_\mathbf{m}))_{s,k} \in \mathrm{SG}\left(\frac{4\gamma^2 d^2(1 + \tau_m)^2}{n}\right), \quad (\nabla\mathcal{L}_{s,m}(\alpha^*_\mathbf{m}))_{t,k} \in \mathrm{SG}\left(\frac{4\gamma^2\tau_m^2}{n}\right).
$$

From the concentration properties of a sub-Gaussian random variable we have:

$$
\mathbb{P}\left(\left|(\nabla\mathcal{L}_{s,m}(\alpha^*_\mathbf{m}))_{s,k} - (\nabla\bar{\mathcal{L}}_{s,m}(\alpha^*_\mathbf{m}))_{s,k}\right| \leq \epsilon\right) \geq 1 - 2\exp\left\{-\frac{n\epsilon^2}{8\gamma^2(1+\tau_m)^2 d^2}\right\}.
$$

and

$$
\mathbb{P}\left(\left|(\nabla\mathcal{L}_{s,m}(\alpha^*_\mathbf{m}))_{t,k} - (\nabla\bar{\mathcal{L}}_{s,m}(\alpha^*_\mathbf{m}))_{t,k}\right| \leq \epsilon\right) \geq 1 - 2\exp\left\{-\frac{n\epsilon^2}{8\gamma^2\tau_m^2}\right\}, \quad \forall t\in V\setminus\{s\}.
$$

Now let $(\nabla\mathcal{L}_{s,m}(\alpha^*_\mathbf{m}))_t = \{(\nabla\mathcal{L}_{s,m}(\alpha^*_\mathbf{m}))_{t,k}\}_{k=1}^{m}$. By union bound we get:

$$
\begin{aligned}
&\mathbb{P}\left(\left\|(\nabla\mathcal{L}_{s,m}(\alpha^*_\mathbf{m}))_s - (\nabla\bar{\mathcal{L}}_{s,m}(\alpha^*_\mathbf{m}))_s\right\|_2 \geq \epsilon\right) \\
&\leq \sum_{k=1}^{m}\mathbb{P}\left(\left|(\nabla\mathcal{L}_{s,m}(\alpha^*_\mathbf{m}))_{s,k} - (\nabla\bar{\mathcal{L}}_{s,m}(\alpha^*_\mathbf{m}))_{s,k}\right| \geq \frac{\epsilon}{\sqrt{m}}\right) \\
&\leq 2m\exp\left\{-\frac{n\epsilon^2}{8\gamma^2(1+\tau_m)^2 m d^2}\right\}
\end{aligned}
$$

and

$$
\mathbb{P}\left(\left\|(\nabla\mathcal{L}_{s,m}(\alpha^*_\mathbf{m}))_t - (\nabla\bar{\mathcal{L}}_{s,m}(\alpha^*_\mathbf{m}))_t\right\|_2 \geq \epsilon\right) \leq 2m\exp\left\{-\frac{n\epsilon^2}{8\gamma^2\tau_m^2 m}\right\}.
$$

By using union bound again we get:

$$
\begin{aligned}
\mathbb{P}\left(\mathcal{R}^*\left(\nabla\mathcal{L}_{s,m}(\alpha^*_\mathbf{m}) - \nabla\bar{\mathcal{L}}_{s,m}(\alpha^*_\mathbf{m})\right) \geq \epsilon\right) &= \mathbb{P}\left(\sup_{t\in V}\left\|(\nabla\mathcal{L}_{s,m}(\alpha^*_\mathbf{m}))_t - (\nabla\bar{\mathcal{L}}_{s,m}(\alpha^*_\mathbf{m}))_t\right\| \geq \epsilon\right) \\
&\leq 2m\exp\left\{-\frac{n\epsilon^2}{8\gamma^2(1+\tau_m)^2 m d^2}\right\} \\
&\quad + 2m(p-1)\exp\left\{-\frac{n\epsilon^2}{8\gamma^2\tau_m^2 m}\right\} \\
&\leq 2mp\exp\left\{-\frac{n\epsilon^2}{8\gamma^2(1+\tau_m)^2 m d^2}\right\}.
\end{aligned}
$$

Choosing $\epsilon = \sqrt{24\gamma^2(1 + \tau_m)^2 m d^2 \log(p)/n}$, we get the following: with probability at least $1 - 2m/p^2$

$$
\mathcal{R}^*\left(\nabla\mathcal{L}_{s,m}(\alpha^*_\mathbf{m}) - \nabla\bar{\mathcal{L}}_{s,m}(\alpha^*_\mathbf{m})\right) \leq \sqrt{24\gamma^2(1 + \tau_m)^2 m d^2 \log(p)/n}.
$$

**(b) Upper bound for $\mathcal{R}^*(\nabla \bar{\mathcal{L}}_{s,m}(\alpha_{\mathbf{m}}^*))$:** From Assumption 2 we have the following upper bound for $\mathcal{R}^*(\nabla \bar{\mathcal{L}}_{s,m}(\alpha_{\mathbf{m}}^*))$:

$$\mathcal{R}^*(\nabla \bar{\mathcal{L}}_{s,m}(\alpha_{\mathbf{m}}^*)) \leq L\mathcal{R}^*(\alpha_{\mathbf{m}}^* - \alpha^*).$$

Combining the results from parts (a) and (b) we get the following: with probability at least $1 - 2m/p^2$

$$\mathcal{R}^*(\nabla \mathcal{L}_{s,m}(\alpha_{\mathbf{m}}^*)) \leq L\mathcal{R}^*(\alpha_{\mathbf{m}}^* - \alpha^*) + c_1 \gamma \tau_m \sqrt{md^2 \log(p)/n},$$

where $c_1 > 0$ is a constant. $\qquad\square$

Following results from Theorem 2, Lemma 1, we conclude that if the regularization parameter $\lambda_n$ is chosen such that

$$\lambda_n \geq 2L\mathcal{R}^*(\alpha_{\mathbf{m}}^* - \alpha^*) + c_1 \gamma \tau_m \sqrt{md^2 \log(p)/n},$$

then with probability at least $1 - 2m/p^2$, any stationary point $\alpha_{\mathbf{m}}$ in $\mathbb{B}_2(\alpha_{\mathbf{m}}^*, r_m)$ satisfies

$$\|\alpha_{\mathbf{m}} - \alpha_{\mathbf{m}}^*\|_2 \leq \frac{6\sqrt{2}}{\kappa}\sqrt{d}\lambda_n.$$

### B.3 Proof of Corollary 4

Let $\alpha_{\mathbf{m}}$ be any stationary point in $\mathbb{B}_2(\alpha_{\mathbf{m}}^*, r_m)$. And suppose $\lambda_n$ is chosen such that:

$$\lambda_n = 2L\mathcal{R}^*(\alpha_{\mathbf{m}}^* - \alpha^*) + c_1 \gamma \tau_m \sqrt{md^2 \log(p)/n}.$$

In this section we derive bounds for the overall estimation error $\|\alpha_{\mathbf{m}} - \alpha^*\|_2$. From triangle inequality, we have:

$$\|\alpha_{\mathbf{m}} - \alpha^*\|_2 \leq \|\alpha_{\mathbf{m}} - \alpha_{\mathbf{m}}^*\|_2 + \|\alpha_{\mathbf{m}}^* - \alpha^*\|_2.$$

From Theorem 2, we have a bound for $\|\alpha_{\mathbf{m}} - \alpha_{\mathbf{m}}^*\|_2$. So, here we focus on bounding $\|\alpha_{\mathbf{m}}^* - \alpha^*\|_2$.

Since the true parameters $B_t^*(.)$ lie in a Sobolev space of order two, we know that there exists a constant $c_2 > 0$ such that [1]:

$$\sup_{t \in V} \|\alpha_{\mathbf{t},\mathbf{m}}^* - \alpha_{\mathbf{t}}^*\|_2 \leq \frac{c_2}{m^2}.$$

Combining this result with the result from Theorem 2, we get, with probability at least $1 - 2m/p^2$:

$$
\begin{aligned}
\|\alpha_{\mathbf{m}} - \alpha^*\|_2 &\leq \|\alpha_{\mathbf{m}} - \alpha_{\mathbf{m}}^*\|_2 + \|\alpha_{\mathbf{m}}^* - \alpha^*\|_2 \\[4pt]
&\leq \frac{6\sqrt{2}}{\kappa}\sqrt{d}\lambda_n + \frac{c_2\sqrt{d}}{m^2} \\[4pt]
&= \frac{6\sqrt{2}}{\kappa}\sqrt{d}\left(2L\mathcal{R}^*(\alpha_{\mathbf{m}}^* - \alpha^*) + c_1\gamma\tau_m\sqrt{md^2\log(p)/n}\right) + \frac{c_2\sqrt{d}}{m^2} \\[4pt]
&\leq \frac{6\sqrt{2}}{\kappa}\sqrt{d}\left(\frac{2c_2 L}{m^2} + c_1\gamma\tau_m\sqrt{md^2\log(p)/n}\right) + \frac{c_2\sqrt{d}}{m^2} \\[4pt]
&\leq c_3\sqrt{d}\left[\frac{L}{\kappa}\frac{1}{m^2} + \gamma\tau_m\sqrt{md^2\log(p)/n}\right],
\end{aligned}
$$

where $c_3 > 0$ is a constant. Choosing $m = \left(\frac{L}{\kappa\gamma\tau_m}\right)^{2/5}\left(\frac{n}{d^2\log(p)}\right)^{1/5}$ gives us the following error bound:

$$\|\alpha_{\mathbf{m}} - \alpha^*\|_2 \leq c_4 \left(\frac{L\gamma^4\tau_m^4}{\kappa}\right)^{1/5}\left(\frac{d^{13/4}\log(p)}{n}\right)^{2/5},$$

and the corresponding $\lambda_n$ for this choice of $m$ is given by:

$$\lambda_n = c_5 \left(\frac{L\gamma^4\tau_m^4}{\kappa}\right)^{1/5}\left(\frac{d^2\log(p)}{n}\right)^{2/5},$$

where $c_4, c_5 > 0$ are constants.

Figure 1: ROC plots for data generated from non-parametric graphical model with $B_s(X) = \sin(4\pi X)$. Top row shows results for chain graph with $C_s(X) = 0$. Bottom row shows results for grid graph with $C_s(X) = 0$.

## C  Experiments

### C.1  Synthetic Data

In this section we present additional results from our synthetic experiments. Specifically, we present ROC plots for $n \in \{100, 200, 500\}$. We use the same parameter settings as described in Section 7 of the main paper, to generate synthetic data. Figure 1 shows ROC plots for data generated from non-parametric graphical model with $B_s(X) = \sin(4\pi X)$. Figure 2 shows ROC plots for $B_s(X) = \left[\exp\left(-20(X - 0.5)^2\right) + \exp\left(-20(X + 0.5)^2\right) - 1\right]$ and Figure 3 shows ROC plots for Gaussian data.

### C.2  Futures Intraday Data

In this section we present the graph learned by GGM for Futures Intraday data and also present more detailed graphs learned by all the three estimators. As pointed out in Section 7, selecting tuning parameter based on held out log likelihood resulted in very dense graphs for Nonparanormal and GGM. So we use a different technique to compare all the models. We fix a tuning parameter for Expxorcist and select tuning parameters for the baselines that resulted in graphs with same number of edges. Figure 4 shows the graph structures learned for one such choice of tuning parameters. Figures 5, 7, 6 present more detailed graphs for the corresponding graphs in Figure 4.

## References

[1] P Bickel, P Diggle, S Feinberg, U Gather, I Olkin, and S Zeger. Springer series in statistics. 2009.

Figure 2: ROC plots for data generated from non-parametric graphical model with $B_s(X) = \left[\exp\left(-20(X-0.5)^2\right) + \exp\left(-20(X+0.5)^2\right) - 1\right]$. Top row shows results for chain graph with $C_s(X) = 0$. Bottom row shows results for grid graph with $C_s(X) = 0$.

Figure 3: ROC plots for data generated from Gaussian distributions. Top row shows plots for chain graph and bottom row shows plots for grid graph.

(a) Nonparanormal

(b) GGM

(c) Expxorcist

Figure 4: Graph Structures learned for the Futures Intraday Data. The Expxorcist graph shown here was obtained by selecting $\lambda = 0.1$. Nodes are colored based on their categories. Edge thickness is proportional to the magnitude of the interaction.

Figure 5: Nonparanormal.

Figure 6: Expxorcist.

Figure 7: GGM.