[Reviews · NeurIPS 2017]

Reviewer 1



This paper proposes a method to estimate a joint multivariate density non-parametrically by estimating a product of marginal conditional exponential densities. The marginal distribution of each variable is conditioned on the neighbors of that variable in a graph. The authors first discuss how their work relate to the literature. They study the consistency of their approach in terms of the estimated marginals with respect to a joint distribution, by relying on a theorem in ref. 28. They also develop an estimation algorithm, study statistical properties of their approach and discuss the relationships with copulas. The paper ends with experimental results. The paper is well written. 1) Are there guarantees if the real distribution does not belong to the family of distribution estimated? 2) I am not very familiar with nonparametric density estimation. How practical is the assumption that the base measures are known? 3) The experiments seem well conducted an the baselines selected in a sensible manner. The evaluation is performed both on synthetic and real data. On synthetic data, the approach is only evaluated on models matching the assumption that the factors are at most of size 2. Imho, testing the approach on more general model would help understand its usefulness. 4) l119: why doesn't nonparametric estimation benefit from estimating a joint distribution as a product of conditional marginal distribution? 5) How easy/hard would it be to extend your approach to cliques of size greater than 2? 6) Will you make your code available? 7) Based on the definition of the density, shouldn't the edges of the graph E mentioned in the introduction correspond to non-independence rather than independence? Typos: You use both "nonparametric" and "non-parametric".